# An Unexpected Reaction of Isodehydracetic Acid with Amines in the Presence of 1-Ethyl-3-(3-dimethylaminopropyl) Carbodiimide Hydrochloride Yields a New Type of β-Enaminones

**DOI:** 10.3390/molecules25092131

**Published:** 2020-05-02

**Authors:** Delong Wang, Hui Shi

**Affiliations:** 1Department of Pharmaceutical Engineering, Shanxi Agricultural University, Taigu 030801, China; 2State Key Laboratory of Coal Conversion, Institute of Coal Chemistry, Chinese Academy of Sciences, Taiyuan 030001, China; shihui@sxicc.ac.cn

**Keywords:** isodehydracetic acid, EDC, ketene

## Abstract

The reaction of isodehydracetic acid with amines was serendipitously found to afford β-enaminones in the presence of the coupling agent 1-ethyl-3-(3-dimethylaminopropyl) carbodiimide hydrochloride (EDC). Under the optimal reaction condition, 23 examples of α-aminomethylene glutaconic anhydride were obtained at approximately 30−80% yields. This is a concise, operationally simple method to expediently synthesize a new type of β-enaminone-containing compound.

## 1. Introduction

β-Enaminones (including β-enaminoesters) can be primely deemed as β-acyl enamines or amides interpolated by an alkene group. These compounds are intrinsically distinct from conventional enamines in reactivity and stability that readily decompose through hydrolytic or oxidative pathways [1,2]. β-Enaminones, however, are quite stable and easily isolated, which may be attributed to the conjugation of the enamine to a carbonyl that attenuates their reactivity. In addition to their distinct reactivity profile, β-enaminones are the important building blocks for a variety of versatile biologically active molecules like indoles [3] and pyrimidines [4], and are significant precursors for β-aminoacids [5]. Optically active β-enaminones were also used to prepare chiral ligands for diastereoselective synthesis [6]. More importantly, they have attracted attention in pharmaceutical applications, such as anticonvulsants [7,8,9] and α7 nicotinic acetylcholine receptor modulators [10]. Their stability under simulated physiological pH and low toxicity are exemplified by their use as orally active medicinal agents [11,12]. A combination of the above factors provides continuous impetus to develop a synthetic route to previously inaccessible or laboriously synthesized *n*-enaminone scaffolds.

Until now, numerous approaches have been developed to construct different types of β-enaminones. The classical approach is the direct condensation of amines with acyclic or cyclic 1,3-diketones or 3-ketoesters, affording the corresponding products (Figure 1a) [13,14,15,16]. When the lactams or lactims reacted with active methylene compounds, β-enaminones with an *N*-heterocycle at the β-site can be prepared (Figure 1b) [13,14]. Others include the methods for the construction of cyclic and α-substituted enaminones (Figure 1c) [17,18,19]. These plentiful scaffolds occurring the β-enaminone unit are pretty meaningful for diversity-oriented synthesis in drug discovery that aims to efficiently generate collections of small molecules with diverse appendages, functional groups, stereochemistry and skeletons, thus yielding diverse biological activities capable of modulating a wide variety of biological process [20,21,22]. We herein present a full disclosure of our serendipitous findings for ready synthesis of a new type of β-enaminone-containing scaffold (Figure 1, **IV**) in a concise and operationally facile way.

## 2. Results and Discussion

Initially, we attempted to synthesize *N*-propyl isodehydracetamide, a hit of succinate dehydrogenase inhibitor in our preliminary virtual screening, by simply treating isodehydracetic acid with *n*-propylamine in the presence of DCC/DMAP (Scheme 1). This reaction was carried out in dry DCM but resulted in a complex mixture which was different from the same reaction employing *n*-propanol instead of *n*-propylamine. Further isolation by column chromatography gave a small number of products. Analysis of its ^13^C-NMR spectrum revealed an apparent abnormal chemical shift changing from 114.5 ppm of isodehydracetic acid to 93.4 ppm. Meanwhile, in the HMBC spectrum (Appendix A), the correlations from -NHCH_2_- to C-6 was observed. This HMBC correlation seemed unreasonable for the anticipated compound 1 (Figure 2). These confusions had accompanied us until isobutylamine was subjected to this reaction. A cubic crystal formed from the pure product solution (50% ethyl acetate/hexane) and the obtained X-ray structure eliminated our existing confusions. As depicted in Figure 3 (data shown in Appendix B), product 3 was a β-enaminone-containing scaffold of α-aminomethylene glutaconic anhydride, rendering a good identity between the NMR data of compound 2 and its structure (Figure 2).

This reaction process was then optimized by systemically varying the reaction conditions (Table 1). As shown from entries 1−3, the EDC/HOBT reaction system was superior over DCC/DMAP and HATU/DIPEA systems. Further optimizations (entries 4 and 5) revealed that this reaction was merely relevant to EDC rather than DIPEA and Et_3_N. In particular, attempts to employ acyl chloride and *n*-propylamine failed to give a separable mixture (entry 10). Notably, decreased yields showed up when the other solvent (entry 7) or a higher temperature (entry 8) was adopted. Increasing the EDC equivalents did not improve the yield, but the reaction time could be shortened to 30 min (entries 4 and 6). These results disclosed that optimized conditions employed EDC for 30 min in DCM at room temperature.

With the optimized reaction condition established, we next explored the scope of this reaction by various amines (Scheme 2). This reaction tolerated a wide range of primary alkyl and aryl amines. As we expected, moderate yields (approximately 30% to 80%) were achieved. Employing alkyl amines (**2**, **3**, and **4a**–**4k**) and electron-rich phenylamines (**4q**–**4t**) furnished a relatively higher yield than electron-deficient phenylamines (**4m**–**4p**). Meanwhile, compared with compounds **2** and **4a**, the bulkier the chains of alkyl amines were (**3** and **4b**–**4e**), the lower the yields of the products obtained, indicating the steric effect in the reaction process. Some exceptions to the generality of this protocol were found during substrate scope studies. As deduced from the NMR spectra, the sterically encumbered tert-butylamine and tert-octylamine gave their corresponding amide products **4u** and **4v**, respectively, as did 2-toluidine (**4w**). 2,6-Dimethylaniline resulted in a complex mixture that contained neither the β-enaminone nor the amide product.

A tentative mechanistic model of the foregoing reaction was formulated on the basis of the following considerations. It has been demonstrated that the presence of a strongly electron-withdrawing group in the α-position to the carboxyl can enable a β-elimination pathway through an E1cB mechanism to afford the corresponding ketene [23,24,25]. These reaction conditions depend closely on the α-substituent and the leaving group in E1cB elimination [24,26], and, obviously, the ketene formation under mild conditions is highly practical. Specifically, the adduct of diethylphosphonoacetic acid and DCC can readily form a ketene intermediate under mild conditions through the E1cB pathway [24]. This ketene serves as an efficient acylation agent for sterically hindered substrates in several minutes. In our case, isodehydracetic acid bears an electron-withdrawing group (-C(CH_3_)=CHCOO-), and, therefore, when it reacts with EDC, a carbodiimide type of dehydrant similar to DCC (a ketene intermediate, **6**) was presumptively formed first (Scheme 3). The electrophilic character of position 6 in particular would be enhanced by the ketene group at position 5, which is thus responsible for the facile acceptance of amine attack and easy elimination of the carboxyl group. This could be verified by a quantum calculation that showed a more positive charge of carbon 6 than that of the ketene carbon (Appendix A). Finally, the leaving carboxyl group would be acylated by ketene, delivering the target molecule **10**. However, the ketene carbon was the preference of a sterically hindered amine, thus giving the amide product **7**, as could be seen from the products **4u**–**4w**.

## 3. Experimental Procedures

### 3.1. General Information

All chemicals were purchased from Macklin Biochemical Co., Ltd. (Shanghai, China) and used without further purification. Anhydrous dichloromethane (DCM) was dispensed from a solvent purification system utilizing calcium hydride. Analytical TLC was performed using precoated plates (silica gel GF254) and visualized with UV light or an I_2_ chamber. Chromatography process was performed on silica gel (200–300 mesh) (Qingdao Haiyang Chemical Co., Ltd., China). A Bruker AM-500 NMR spectrometer (Bruker, Germany) was used for 1D and 2D NMR tests and the chemical shifts (*δ* expressed in *ppm* were refered to the solvent (DMSO-*d*_6_) residual peaks (*δ*_H_ 2.54/ *δ*_C_ 40.45). Multiplicities are given as s (singlet), d (doublet), t (triplet), q (quartet), and m (multiplet). Coupling constants (*J*) are given in Hz. The HRESIMS data were collected on Waters Xevo G2-XS TOF mass spectrometer (Waters Co., Milford, MA, USA). Uncorrected melting point values were gathered using WRS-1B Digital Melting-point Apparatus (Shanghai JiaHang Instruments).

### 3.2. Synthesis of Isodehydracetic Acid



To a 2 L three-necked flask fitted with a thermometer, a stirrer, and a dropping funnel, 450 mL of concentrated sulfuric acid was added. This flask was cooled in an ice bath for 10 min, and 325 g (2.5 moles) ethyl acetoacetate was added to the above stirring acid at such a rate that the temperature remains between 10 °C and 15 °C. When all the ester had been added, the flask was stoppered with a calcium chloride drying tube and allowed to stand at 20 °C. After 4 days, the reaction mixture was poured onto 1000 g of crushed ice while being stirred vigorously with a wooden paddle. The resulting solid was collected on a large Büchner funnel, washed with two 200 mL portions of cold water, and sucked as dry as possible. The sucked solid was then crushed and naturally air dried for a week. The dried solid was subjected to a silica gel column eluting with petroleum ether/ethyl acetate (1:5) to afford crude isodehydracetic acid. The crude product was dissolved in hot ethyl acetate and cooled slowly to effect crystallization. The yield of isodehydroacetic acid was approximately 30 g, white solid, m.p. 154–155 °C; ^1^H-NMR (500 MHz, DMSO-*d*_6_) δ 13.53 (s, 1H), 6.16 (s, 1H), 2.38 (s, 3H), 2.22 (s, 3H); ^13^C-NMR (125 MHz, DMSO-*d*_6_) δ 167.4, 164.3, 160.8, 155.6, 114.5, 112.0, 21.4, 20.0. ESI-MS *m*/*z*: 169.2 [M + H]^+^ [27].

### 3.3. General Procedure for Synthesis of Compounds ***2, 3 and 4a**–**4w***



A round-bottomed flask was successively charged with isodehydracetic acid (1.0 mmol), 1-(3-dimethylaminopropyl)-3-ethylcarbodiimide hydrochloride (EDC) (1.2 mmol) and DCM (5 mL). The mixture was stirred at room temperature for 10 min. After that, amine (1.2 mmol) was slowly added. The system was reacted for 30 min. Once complete, the mixture was washed with 3 × 5 mL distilled water and the organic phase was dried with anhydrous Na_2_SO_4_ and concentrated. The residual was subjected to a silica gel column eluting with petroleum ether/ethyl acetate (from 5:0 to 5:1, 5:3, and 5:5) to afford a product. The product was recrystallized from petroleum ether/ethyl acetate if necessary.

*(Z)-4-Methyl-3-(1-(propylamino)ethylidene)-2H-pyran-2,6(3H)-dione* (**2**): colorless crystal, m.p. 56–58 °C, yield 66%; ^1^H-NMR (500 MHz, DMSO-*d*_6_) δ 11.74 (s, 1H), 5.44 (s, 1H), 3.51–3.53 (m, 2H), 2.51 (s, 3H), 2.35 (s, 3H), 1.66–1.70 (m, 2H), 0.99 (t, *J* = 7.2 Hz, 3H); ^13^C-NMR (125 MHz, DMSO-*d*_6_) δ 172.2, 166.7, 161.3, 158.9, 103.8, 94.8, 46.5, 25.5, 22.8, 20.0, 11.9. HRMS (ESI) calcd. for C_11_H_16_NO_3_ [M + H]^+^ 210.1125, found 210.1129.

*(Z)-3-(1-(Isopropylamino)ethylidene)-4-methyl-2H-pyran-2,6(3H)-dione* (**4a**): colorless crystal, m.p. 62–64 °C, yield 57%; ^1^H-NMR (500 MHz, DMSO-*d*_6_) δ 11.70 (s, 1H), 5.41 (s, 1H), 4.09–4.17 (m, 1H), 2.51 (s, 3H), 2.31 (s, 3H), 1.27 (s, 3H), 1.26 (s, 3H); ^13^C-NMR (125 MHz, DMSO-*d*_6_) δ 171.3, 166.9, 161.5, 159.2, 103.8, 94.7, 47.1, 25.6, 23.5, 20.0. HRMS (ESI) calcd. for C_11_H_16_NO_3_ [M + H]^+^ 210.1125, found 210.1129.

*(Z)-3-(1-(Isobutylamino)ethylidene)-4-methyl-2H-pyran-2,6(3H)-dione* (**3**): colorless crystal, m.p. 74–76 °C, yield 52%; ^1^H-NMR (500 MHz, DMSO-*d*_6_) δ 11.86 (s, 1H), 5.45 (s, 1H), 3.39–3.42 (m, 2H), 2.50 (s, 3H), 2.35 (s, 3H), 1.90–2.00 (m, 1H), 1.00 (d, *J* = 6.7 Hz, 6H); ^13^C-NMR (125 MHz, DMSO-*d*_6_) δ 172.4, 167.1, 161.4, 159.1, 104.0, 94.9, 52.1, 28.8, 25.7, 20.7, 20.2. HRMS (ESI) calcd. for C_12_H_18_NO_3_ [M + H]^+^ 224.1281, found 224.1285.

*(Z)-4-Methyl-3-(1-(octylamino)ethylidene)-2H-pyran-2,6(3H)-dione* (**4b**): white solid, m.p. 110–112 °C, yield 51%; ^1^H-NMR (500 MHz, DMSO-*d*_6_) δ 11.72 (s, 1H), 5.44 (s, 1H), 3.52–3.54 (m, 2H), 2.50 (s, 3H), 2.34 (s, 3H), 1.61–1.68 (m, 2H), 1.29–1.38 (m, 10H), 0.89 (t, *J* = 7.1 Hz, 3H); ^13^C-NMR (125 MHz, DMSO-*d*_6_) δ 172.4, 166.9, 161.5, 159.1, 103.8, 94.9, 45.0, 32.2, 29.5, 29.5, 29.4, 27.1, 25.6, 23.1, 20.1, 15.0. HRMS (ESI) calcd. for C_16_H_26_NO_3_ [M + H]^+^ 280.1907, found 280.1912.

*(Z)-3-(1-(Dodecylamino)ethylidene)-4-methyl-2H-pyran-2,6(3H)-dione* (**4c**): white solid, m.p. 97–99 °C, yield 47%; ^1^H-NMR (500 MHz, DMSO-*d*_6_) δ 11.72 (s, 1H), 5.44 (s, 1H), 3.50–3.54 (m, 2H), 2.50 (s, 3H), 2.34 (s, 3H), 1.60–1.67 (m, 2H), 1.21–1.37 (m, 18H), 0.88 (t, *J* = 7.0 Hz, 3H); ^13^C-NMR (125 MHz, DMSO-*d*_6_) δ 172.3, 166.9, 161.5, 159.1, 103.8, 94.9, 45.0, 32.3, 30.0, 29.9, 29.9, 29.7, 29.5, 29.4, 27.1, 25.6, 23.1, 20.1, 15.0. HRMS (ESI) calcd. for C_20_H_34_NO_3_ [M + H]^+^ 36.2533, found 36.2539.

*(Z)-3-(1-(Hexadecylamino)ethylidene)-4-methyl-2H-pyran-2,6(3H)-dione* (**4d**): white solid, m.p. 109–112 °C, yield 56%; ^1^H-NMR (500 MHz, DMSO-*d*_6_) δ 11.72 (s, 1H), 5.44 (s, 1H), 3.51–3.54 (m, 2H), 2.51 (s, 3H), 2.35 (s, 3H), 1.63–1.69 (m, 2H), 1.21–1.37 (m, 26H), 0.89 (t, *J* = 7.0 Hz, 3H); ^13^C-NMR (125 MHz, DMSO-*d*_6_) δ 171.6, 166.4, 160.8, 158.3, 103.5, 94.4, 44.5, 31.7, 29.5, 29.4, 29.4, 29.3, 29.1, 29.0, 28.9, 26.6, 25.0, 22.5, 19.5, 14.3. HRMS (ESI) calcd. for C_24_H_42_NO_3_ [M + H]^+^ 392.3159, found 392.3165.

*(Z)-3-(1-((2-Ethylhexyl)amino)ethylidene)-4-methyl-2H-pyran-2,6(3H)-dione* (**4e**): colorless crystal, m.p. 95–96 °C, yield 43%; ^1^H-NMR (500 MHz, DMSO-*d*_6_) δ 11.87 (s, 1H), 5.45 (s, 1H), 3.48–3.50 (m, 2H), 2.52 (s, 3H), 2.36 (s, 3H), 1.63–1.66 (m, 1H), 1.31–1.43 (m, 8H), 0.91–0.993 (overlap, 6H); ^13^C-NMR (125 MHz, DMSO-*d*_6_) δ 172.4, 167.1, 161.2, 158.9, 103.9, 94.9, 47.8, 39.3, 31.2, 29.0, 25.6, 24.6, 23.3, 20.1, 14.8, 11.5. HRMS (ESI) calcd. for C_16_H_26_NO_3_ [M + H]^+^ 280.1907, found 280.1914.

*(Z)-3-(1-(Cyclopropylamino)ethylidene)-4-methyl-2H-pyran-2,6(3H)-dione* (**4f**): white solid, m.p. 77–79 °C, yield 71%; ^1^H-NMR (500 MHz, DMSO-*d*_6_) δ 11.64 (s, 1H), 5.48 (s, 1H), 3.03–3.06 (m, 1H), 2.67 (s, 3H), 2.36 (s, 3H), 0.97–1.01 (m, 2H), 0.80–0.84 (m, 2H); ^13^C-NMR (125 MHz, DMSO-*d*_6_) δ 174.5, 166.7, 161.1, 158.8, 104.4, 94.9, 27.2, 25.5, 21.0, 8.4. HRMS (ESI) calcd. for C_11_H_14_NO_3_ [M + H]^+^ 208.0968, found 208.0974.

*(Z)-3-(1-(Cyclopentylamino)ethylidene)-4-methyl-2H-pyran-2,6(3H)-dione* (**4g**): white solid, m.p. 105–107 °C, yield 69%; ^1^H-NMR (500 MHz, DMSO-*d*_6_) δ 11.91 (s, 1H), 5.45 (s, 1H), 4.31–4.33 (m, 1H), 2.55 (s, 3H), 2.35 (s, 3H), 2.02–2.08 (m, 2H), 1.61–1.75 (m, 6H); ^13^C-NMR (125 MHz, DMSO-*d*_6_) δ 171.5, 166.9, 161.3, 159.0, 103.8, 94.8, 56.3, 33.9, 25.5, 24.4, 20.5. HRMS (ESI) calcd. for C_13_H_18_NO_3_ [M + H]^+^ 236.1281, found 236.1289.

*(Z)-3-(1-(Cyclohexylamino)ethylidene)-4-methyl-2H-pyran-2,6(3H)-dione* (**4h**): white solid, m.p. 122–124 °C, yield 65%; ^1^H-NMR (500 MHz, DMSO-*d*_6_) δ 11.90 (s, 1H), 5.45 (s, 1H), 3.87–3.93 (m, 1H), 2.55 (s, 3H), 2.36 (s, 3H), 1.31–1.92 (m, 10H); ^13^C-NMR (125 MHz, DMSO-*d*_6_) δ 171.0, 166.9, 161.2, 159.0, 103.9, 94.7, 53.2, 33.0, 25.6, 25.5, 24.4, 19.8. HRMS (ESI) calcd. for C_14_H_20_NO_3_ [M + H]^+^ 250.1438, found 250.1443.

*(Z)-3-(1-(Benzylamino)ethylidene)-4-methyl-2H-pyran-2,6(3H)-dione* (**4i**): white solid, yield 65%; ^1^H-NMR (500 MHz, DMSO-*d*_6_) δ 11.99 (s, 1H), 7.35–7.46 (m, 5H), 5.48 (s, 1H), 4.83 (s, 2H), 2.52 (s, 3H), 2.35 (s, 3H); ^13^C-NMR (125 MHz, DMSO-*d*_6_) δ 172.2, 166.7, 161.2, 158.9, 137.3, 129.8, 128.7, 128.4, 104.4, 95.2, 48.4, 25.6, 20.3. HRMS (ESI) calcd. for C_15_H_16_NO_3_ [M + H]^+^ 258.1125, found 258.1129.

*(Z)-4-Methyl-3-(1-(phenethylamino)ethylidene)-2H-pyran-2,6(3H)-dione* (**4j**): white solid, yield 73%; ^1^H-NMR (500 MHz, DMSO-*d*_6_) δ 11.72 (s, 1H), 7.26–7.37 (m, 5H), 5.44 (s, 1H), 3.82 (t, *J* = 7.0 Hz, 2H), 2.98 (t, *J* = 7.0 Hz, 2H), 2.43 (s, 3H), 2.30 (s, 3H); ^13^C-NMR (125 MHz, DMSO-*d*_6_) δ 172.1, 166.6, 161.3, 158.8, 138.9 129.8, 129.4, 127.5, 104.0, 94.9, 46.3, 35.5, 25.5, 20.0. HRMS (ESI) calcd. for C_16_H_18_NO_3_ [M + H]^+^ 272.1281, found 272.1281.

*(Z)-3-(1-((Furan-2-ylmethyl)amino)ethylidene)-4-methyl-2H-pyran-2,6(3H)-dione* (**4k**): white solid, yield 55%; ^1^H-NMR (500 MHz, DMSO-*d*_6_) δ 11.94 (s, 1H), 7.73 (s, 1H), 6.51 (m, 2H), 5.55 (s, 1H), 4.87 (d, *J* = 5.0 Hz, 2H), 3.82 (t, *J* = 7.0 Hz, 2H), 2.98 (t, *J* = 7.0 Hz, 2H), 2.60 (s, 3H), 2.36 (s, 3H); ^13^C-NMR (125 MHz, DMSO-*d*_6_) δ 172.2, 166.8, 161.1, 158.9, 150.1, 144.4, 111.7, 109.6, 104.7, 95.2, 41.6, 25.6, 20.1. HRMS (ESI) calcd. for C_13_H_14_NO_4_ [M + H]^+^ 248.0917, found 248.0921.

*(Z)-4-Methyl-3-(1-(phenylamino)ethylidene)-2H-pyran-2,6(3H)-dione* (**4l**): white solid, m.p. 143–144 °C, yield 72%; ^1^H-NMR (500 MHz, DMSO-*d*_6_) δ 13.34 (s, 1H), 7.39–7.57 (m, 5H), 5.63 (s, 1H), 2.51 (s, 3H), 2.43 (s, 3H); ^13^C-NMR (125 MHz, DMSO-*d*_6_) δ 170.6, 167.0, 160.8, 159.0, 137.3, 130.5, 128.8, 126.8, 105.6, 96.2, 25.5, 21.8. HRMS (ESI) calcd. for C_14_H_13_NNaO_3_ [M + Na]^+^ 266.0788, found 266.0792.

*(Z)-3-(1-((4-Fluorophenyl)amino)ethylidene)-4-methyl-2H-pyran-2,6(3H)-dione* (**4m**): white solid, m.p. 137–139 °C, yield 50%; ^1^H-NMR (500 MHz, DMSO-*d*_6_) δ 13.22 (s, 1H), 7.36–7.47 (m, 4H), 5.63 (s, 1H), 2.46 (s, 3H), 2.42 (s, 3H); ^13^C-NMR (125 MHz, DMSO-*d*_6_) δ 171.0, 167.0, 161.0, 159.2, 152.4 (d, *J*_C-F_ = 254.2 Hz), 133.7 (d, *J*_C-F_ = 3.4 Hz), 129.2 (d, *J*_C-F_ = 8.7 Hz), 117.4 (d, *J*_C-F_ = 23.4 Hz), 105.7, 96.3, 25.6, 21.9. HRMS (ESI) calcd. for C_14_H_12_FNNaO_3_ [M + Na]^+^ 284.0693, found 284.0695.

*(Z)-3-(1-((4-Bromophenyl)amino)ethylidene)-4-methyl-2H-pyran-2,6(3H)-dione* (**4n**): white solid, m.p. 161–164 °C, yield 53%; ^1^H-NMR (500 MHz, DMSO-*d*_6_) δ 13.21 (s, 1H), 7.74 (d, *J* = 8.6 Hz, 2H), 7.36 (d, *J* = 8.6 Hz, 2H), 5.65 (s, 1H), 2.49 (s, 3H), 2.42 (s, 3H); ^13^C-NMR (125 MHz, DMSO-*d*_6_) δ 170.4, 166.9, 160.8, 158.9, 136.8, 133.3, 129.0, 121.6, 105.9, 96.6, 25.5, 21.9. HRMS (ESI) calcd. for C_14_H_12_BrNNaO_3_ [M + Na]^+^ 343.9893, found 343.9897.

*(Z)-3-(1-((3,5-dichlorophenyl)amino)ethylidene)-4-methyl-2H-pyran-2,6(3H)-dione* (**4o**): white solid, m.p. 166–167 °C, yield 41%; ^1^H-NMR (500 MHz, DMSO-*d*_6_) δ 13.11 (s, 1H), 7.58–7.70 (m, 3H), 5.68 (s, 1H), 2.51 (s, 3H), 2.42 (s, 3H); ^13^C-NMR (125 MHz, DMSO-*d*_6_) δ 170.4, 166.7, 160.7, 158.7, 139.9, 135.4, 128.3, 126.1, 106.5, 97.1, 25.5, 22.2. HRMS (ESI) calcd. for C_14_H_11_Cl_2_NNaO_3_ [M + Na]^+^ 334.0008, found 334.0012.

*(Z)-3-(1-((4-Chloro-3-(trifluoromethyl)phenyl)amino)ethylidene)-4-methyl-2H-pyran-2,6(3H)-dione* (**4p**): white solid, m.p. 109–111 °C, yield 39%; ^1^H-NMR (500 MHz, DMSO-*d*_6_) δ 13.14 (s, 1H), 7.91 (s, 1H), 7.87 (d, *J* = 8.6 Hz, 2H), 7.73 (d, *J* = 8.6 Hz, 2H), 5.67 (s, 1H), 2.50 (s, 3H), 2.43 (s, 3H); ^13^C-NMR (125 MHz, DMSO-*d*_6_) δ 170.1, 166.5, 160.5, 158.3, 136.9, 133.4, 132.4, 130.3, 128.2 (q, *J*_C-F_ = 30.9 Hz), 126.5 (q, *J*_C-F_ = 5.3 Hz), 123.2 (q, *J*_C-F_ = 274.2 Hz), 106.4, 96.9, 25.2, 21.8. HRMS (ESI) calcd. for C_15_H_11_ClF_3_NNaO_3_ [M + Na]^+^ 368.0272, found 368.0272.

*(Z)-4-Methyl-3-(1-(p-tolylamino)ethylidene)-2H-pyran-2,6(3H)-dione* (**4q**): white solid, m.p. 127–129 °C, yield 75%; ^1^H-NMR (500 MHz, DMSO-*d*_6_) δ 13.29 (s, 1H), 7.35 (d, *J* = 8.0 Hz, 3H), 7.27 (d, *J* = 8.0 Hz, 3H), 5.62 (s, 1H), 2.49 (s, 3H), 2.42 (s, 3H), 2.39 (s, 3H); ^13^C-NMR (125 MHz, DMSO-*d*_6_) δ 170.6, 167.0, 160.9, 159.0, 138.4, 134.7, 130.9, 126.6, 105.5, 96.1, 25.5, 21.8, 21.5. HRMS (ESI) calcd. for C_15_H_15_NNaO_3_ [M + Na]^+^ 280.0944, found 280.0946.

*(Z)-3-(1-((4-Isopropylphenyl)amino)ethylidene)-4-methyl-2H-pyran-2,6(3H)-dione* (**4r**): white solid, m.p. 134–136 °C, yield 70%; ^1^H-NMR (500 MHz, DMSO-*d*_6_) δ 13.30 (s, 1H), 7.42 (d, *J* = 8.0 Hz, 3H), 7.30 (d, *J* = 8.0 Hz, 3H), 5.62 (s, 1H), 2.96–3.01 (m, 1H), 2.50 (s, 3H), 2.42 (s, 3H), 1.24 (d, *J* = 6.9 Hz, 6H); ^13^C-NMR (125 MHz, DMSO-*d*_6_) δ 170.6, 167.0, 160.9, 159.0, 149.1, 135.0, 128.3, 126.6, 105.4, 96.1, 33.9, 25.5, 24.6, 21.8. HRMS (ESI) calcd. for C_17_H_19_NNaO_3_ [M + Na]^+^ 308.1257, found 308.1257.

*(Z)-3-(1-((4-Methoxyphenyl)amino)ethylidene)-4-methyl-2H-pyran-2,6(3H)-dione* (**4s**): white solid, m.p. 148–150 °C, yield 71%; ^1^H-NMR (500 MHz, DMSO-*d*_6_) δ 13.22 (s, 1H), 7.32 (d, *J* = 8.0 Hz, 3H), 7.09 (d, *J* = 8.0 Hz, 3H), 5.60 (s, 1H), 3.84 (s, 3H), 2.47 (s, 3H), 2.42 (s, 3H); ^13^C-NMR (125 MHz, DMSO-*d*_6_) δ 170.8, 167.0, 160.9, 159.6, 159.0, 129.9, 128.1, 115.6, 105.3, 95.9, 56.4, 25.5, 21.7. HRMS (ESI) calcd. for C_15_H_15_NNaO_4_ [M + Na]^+^ 296.0893, found 296.0895.

*(Z)-4-Methyl-3-(1-((4-(trifluoromethoxy)phenyl)amino)ethylidene)-2H-pyran-2,6(3H)-dione* (**4t**): white solid, m.p. 146–147 °C, yield 57%; ^1^H-NMR (500 MHz, DMSO-*d*_6_) δ 13.28 (s, 1H), 7.55 (s, 4H), 5.66 (s, 1H), 2.50 (s, 3H), 2.43 (s, 3H); ^13^C-NMR (125 MHz, DMSO-*d*_6_) δ 170.6, 166.9, 160.8, 158.9, 148.2, 136.5, 129.0, 123.0, 120.9 (q, *J*_C-F_ = 260.7 Hz), 106.0, 96.6, 25.5, 21.9. HRMS (ESI) calcd. for C_15_H_12_F3NNaO_4_ [M + Na]^+^ 350.0611, found 350.0612.

*(Z)-3-(1-(Tert-butylamino)ethylidene)-4-methyl-2H-pyran-2,6(3H)-dione* (**4u**): white solid, m.p. 133–135 °C, yield 44%; ^1^H-NMR (500 MHz, DMSO-*d*_6_) δ 8.18 (s, 1H), 6.10 (s, 1H), 2.20 (s, 3H), 2.09 (s, 3H), 1.35 (s, 9H); ^13^C-NMR (125 MHz, DMSO-*d*_6_) δ 163.9, 160.8, 158.2, 154.8, 118.0, 110.3, 50.8, 28.1, 18.7, 17.7. HRMS (ESI) calcd. for C_12_H_18_NO_3_ [M + H]^+^ 224.1281, found 224.1281.

*4,6-Dimethyl-2-oxo-N-(2,4,4-trimethylpentan-2-yl)-2H-pyran-5-carboxamide* (**4v**): white solid, m.p. 129–131 °C, yield 47%; ^1^H-NMR (500 MHz, DMSO-*d*_6_) δ 7.98 (s, 1H), 6.10 (s, 1H), 2.23 (s, 3H), 2.12 (s, 3H), 1.78 (s, 2H), 1.42 (s, 6H), 1.02 (s, 9H); ^13^C-NMR (125 MHz, DMSO-*d*_6_) δ 164.6, 161.6, 159.5, 155.8, 119.0, 111.3, 56.1, 52.1, 32.2, 32.1, 29.2, 20.0, 18.9. HRMS (ESI) calcd. for C_16_H_26_NO_3_ [M + H]^+^ 280.1907, found 280.1909.

*4,6-Dimethyl-2-oxo-N-(o-tolyl)-2H-pyran-5-carboxamide* (**4w**): white solid, m.p. 138–139 °C, yield 37%; ^1^H-NMR (500 MHz, DMSO-*d*_6_) δ 10.03 (s, 1H), 7.18–7.47 (m, 4H), 6.22 (s, 1H), 2.37 (s, 3H), 2.29 (s, 3H), 2.23 (s, 3H); ^13^C-NMR (125 MHz, DMSO-*d*_6_) δ 164.2, 161.5, 160.0, 155.5, 136.3, 133.4, 131.4, 127.1, 127.0, 126.5, 118.3, 111.5, 20.1, 19.1, 19.0. HRMS (ESI) calcd. for C_15_H_16_NO_3_ [M + H]^+^ 258.1125, found 258.1129.

## 4. Conclusions

In summary, owing to an unexpected reaction, a concise, operationally simple method has been developed which provides expedient access to synthesize a new type of β-enaminone-containing compound. The unique structure of isodehydracetic acid strongly suggests that the reaction mechanism involves the formation of highly reactive ketene intermediates.

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
