# Peer review of "An Unexpected Reaction of Isodehydracetic Acid with Amines in the Presence of 1-Ethyl-3-(3-dimethylaminopropyl) Carbodiimide Hydrochloride Yields a New Type of β-Enaminones"

_molecules, 2020, doi:10.3390/molecules25092131_

Round 1
Reviewer 1 Report
The work by D. Wang and H. Shi describes the serendipitous discovery of the reaction between isodehydracetic acid with amines providing a series of b-enaminones, in the presence of the coupling agent EDC. A plausible mechanism has been proposed by the authors that are centred on the formation of a highly electrophilic ketene intermediate that triggers a peculiar ring-opening ring-closure sequence to yield the enaminone products in acceptable yields.
In my opinion the discovered reaction is quite interesting and may be of use in both an organic and medicinal context. A series of major points though must be faced for the paper to be accepted for publication in Molecules.
Major points:
- The introduction is not clear. Too many different mechanisms are described without the corresponding graphics. For example in Page 1, Line 36:”The most classical among them is direct condensation of amines with acyclic or cyclic 1” but up to this point no molecules have been numbered as 1. I suggest to improve Figure 1, including also the main mechanist pathways involved in the formation of enaminones.
- About Scheme 1. Each molecule must be numbered and so has to be cited within the text. Blu arrows in the products are misleading and unclear. I suggest adding a new Figure describing HMBC contacts.
- About Figure 2. I suggest to add the structure related to the ORTEP close to it for clarity
- About Table 1. A graphical general reaction should be added at the top of the Table for clarity as in Scheme 2. Here, if the yields refer to isolated yields, it should be indicated.
- About Scheme 3. All molecules must be numbered, and the related mechanism description should be improved.
- English style and syntax should be improved;
Overall, this reviewer retains that this manuscript may be worth publication in Molecules provided that all the indicated points are clearly addressed.
Reviewer 2 Report
Wang and Shi made a really good work,
The research is concise and well presented, the results are surely worth of publication in molecules.
The only comment I can add is that the authors need to remove the text form line 117 to 122. The state of the art is properly referenced, the experimental design is clear, the results are well presented and the conclusions fit the content and concise.
Reviewer 3 Report
The submitted paper describes the unexpected results obtained by the reaction between isohydracetic acid and various amines. the reaction conditions were optimized to provide a library of about 20 enaminone derivatives, a family of compounds that have already been studied for various therapeutic applications.
The reaction mechanisms are proposed via highly reactive intermediate ketenes. They are discussed and seem relevant.
Apart from the results that seem interesting to publish it appears that the manuscript does not correspond to the journal's requirements regarding the quality of the English language.
In addition, authors have forgotten parts of the initial template (especially in their conclusion), references are not in the right style please check.
The journal's names not abbreviated as it should be done. the style of the experimental part with a description of the product's data is not in the style usually demanded, please check product's names (in italic ?)
Round 2
Reviewer 1 Report
I've read with interest the revised version of the paper by Wang and Shi about the serendipitous discovery of the reaction between isodehydracetic acid with amines in the presence of EDC, providing a series of beta-enaminones. In my opinion the paper has been improved to a sufficient level of quality and novelty to be published in Molecules.